# The relation between local and distal muscle fat infiltration in chronic whiplash using magnetic resonance imaging

Anette Karlsson[1,2]*, Anneli Peolsson[2,3], James Elliott[4,5], Thobias Romu[1,2], Helena Ljunggren[3], Magnus Borga[1,2], Olof Dahlqvist Leinhard[2,6]

1 Department of Biomedical Engineering, Linköping University, Linköping, Sweden, 2 Center for Medical Image Science and Visualization, Linköping University, Linköping, Sweden, 3 Department of Medical and Health Sciences, Physiotherapy, Linköping University, Linköping, Sweden, 4 Faculty of Health Sciences, The University of Sydney, Northern Sydney Local Health District, The Kolling Institute, St Leonards, NSW, Australia, 5 Department of Physical Therapy and Human Movement Sciences, Feinberg School of Medicine, Northwestern University, Chicago, IL, United States of America, 6 Department of Medical and Health Sciences, Linköping University, Linköping, Sweden

* anette.k.karlsson@liu.se

**Data Availability Statement:** All relevant data are within the manuscript and its Supporting Information files.

**Funding:** The study received funding from the Swedish Research Council and the Medical

## Abstract

The objective of this study was to investigate the relationship between fat infiltration in the cervical multifidi and fat infiltration measured in the lower extremities to move further into understanding the complex signs and symptoms arising from a whiplash trauma. Thirty-one individuals with chronic whiplash associated disorders, stratified into a mild/moderate group and a severe group, together with 31 age- and gender matched controls were enrolled in this study. Magnetic resonance imaging was used to acquire a 3D volume of the neck and of the whole-body. Cervical multifidi was used to represent muscles local to the whiplash trauma and all muscles below the hip joint, the lower extremities, were representing wide-spread muscles distal to the site of the trauma. The fat infiltration was determined by fat fraction in the segmented images. There was a linear correlation between local and distal muscle fat infiltration (p<0.001, $r^2 = 0.28$). The correlation remained significant when adjusting for age and WAD group (p = 0.009) as well as when correcting for age, WAD group and BMI (p = 0.002). There was a correlation between local and distal muscle fat infiltration within the severe WAD group (p = 0.0016, $r^2 = 0.69$) and in the healthy group (p = 0.022, $r^2 = 0.17$) but not in the mild/moderate group (p = 0.29, $r^2 = 0.06$). No significant differences (p = 0.11) in the lower extremities' MFI between the different groups were found. The absence of differences between the groups in terms of lower extremities' muscle fat infiltration indicates that, in this particular population, the whiplash trauma has a local effect on muscle fat infiltration rather than a generalized.

## Introduction

Approximately 50% of individuals involved in a motor vehicle collision should expect to demonstrate signs of recovery within the first 6–12 weeks following the collision event [1]. Others

Research Council of South-East Sweden (FORSS). The funders had no role in study design, data collection and analysis, decision to publish, or preparation of the manuscript. Furthermore, the commercial company AMRA Medical AB provided support in the form of salaries for authors TR, MB and ODL, but did not have any additional role in the study design, data collection and analysis, decision to publish, or preparation of the manuscript. The specific roles of these authors are articulated in the 'author contributions' section.

**Competing interests:** We have read the journal's policy and the authors of this manuscript have the following competing interests: TR, MB, and ODL receive salaries and are stockholders of AMRA Medical AB. AK is a stockholder of AMRA Medical AB. This does not alter our adherence to PLOS ONE's policies on sharing data and material.

will transition from acute to chronic trauma-related disability, presenting with physical and psychological symptoms including neck pain, radiating arm pain, headache, anxiety and depression [1–3]. However, there exists no standard by which to objectively diagnose whiplash associated disorders (WAD), rather the grading is reliant on physical examination findings [4] and a history of a whiplash from a motor vehicle collision.

Recently, cross-sectional and longitudinal studies across three different countries (and insurance schemas) have qualified and quantified larger magnitudes of muscle fat infiltration (MFI) in the cervical multifidi of participants with persistent WAD compared to those reporting lower levels of pain-related disability, those nominating full recovery, idiopathic neck pain, and healthy controls [5–7]. The cervical multifidi may be directly involved during the whiplash trauma as the muscles are inserted directly to the facet capsules of the cervical vertebrae [8]. A neck injury due to the whiplash trauma may explain the higher amount of MFI in multifidi of patients with chronic whiplash [9], although, the mechanism behind MFI reported in cervical multifidi [5–7] is unclear. A number of hypotheses around injury severity, pain intensity and/ or related disability [10, 11], heightened stress-responses [12–14] central/peripheral neuronal interference [15–17], spinal cord injury [18] and/or physical inactivity [19, 20] may be offered to explain the rapid expression and larger magnitude of MFI in patients with more severe self-reported symptoms. Factors such as heightened stress response and physical inactivity are suggested to have general impact on muscle fat infiltration [21], but local injuries would only have an impact on the neck.

Furthermore, a previous review of WAD [21] highlighted the potential influence of and main actions exerted by the sympathetic nervous system on widespread motor function with a focus on the underlying mechanisms for the onset and maintenance of chronic pain. While much of the subsequent research has provided breakthrough knowledge about WAD related pain-processing deficits and psychological distress, the landscape of available literature focusing on the muscle system and motor output following whiplash continues to expand, but not yet in parallel. Albeit preliminary, a case-study reported MFI in both the neck region and the lower extremities [15] that could point to a mechanical injury involving descending white matter pathways of the cervical spinal cord. A higher value of MFI detected in both the lower extremity and neck musculature corresponded to altered spinal cord anatomy and reductions in the ability to maximally activate plantar flexor torques [15]. The preliminary nature of the case-based study [15] supports the need for a larger cohort to further investigate the potential link between MFI in muscles directly associated with the trauma likely affecting the head and neck and in muscles distal to the site of potential injury.

Contrary to the findings by the case study above, a study by Pedler et al, on a larger cohort, found no differences in fat infiltration in the right soleus muscle on group level where the groups were divided into moderate/severe WAD, mild WAD and healthy controls [9]. These contrasting findings calls for further investigations of local and distal MFI in WAD patients. The study by Pedler et al. only investigated one muscle in the lower extremities. In addition, no regression models were used to compare the associations between the local and distal fat compartments.

The aim of this study was to investigate the potential pathophysiological link between local MFI (that may have been injured in the trauma) and generalized MFI distal to the whiplash trauma in patients with severe WAD, in those with mild/moderate WAD, and in healthy controls.

## Method

### Participants

Thirty-one individuals with mild/moderate (N = 20) or severe (N = 11) chronic WAD (at least 6 months duration) and 31 healthy matched controls were included. The Neck Disability

**Table 1. Descriptive data in format: [mean] ± [standard deviation] ([Range]).**

| | Healthy Controls | WAD | | |
|---|---|---|---|---|
| | Total | Total | NDI<40% | NDI≥40% |
| Participants, n | 31 | 31 | 20 | 11 |
| Age, y | 41.5 ± 0.6 (22–61) | 41.5 ± 0.9 (20–62) | 39.2 ± 11.5 (20–62) | 45.7 ± 8.5 (34–58) |
| Body mass index, kg/m$^2$ | 24.4 ± 3.2 (19.7–34.5) | 25.6 ± 4.1 (19.1–33.8) | 25.5 ± 4.1 (19.1–33.8) | 25.8 ± 3.4 (20.3–32.3) |
| NDI, % | N/A | 35.8 ± 14.1 (10–68) | 27.3 ± 6.8 (10–38) | 51.3 ± 10.2 (40–68) |
| Time since injury, mo | N/A | 18.1 ± 9.2 (6–36) | 20.1 ± 9.8 (7–36) | 14.5 ± 7.2 (6–32) |

WAD: Whiplash Associated Disorders, NDI: Neck Disability IndexBMI: Body Mass Index kg/m$^2$

Index (NDI) was used for stratification (mild/moderate: 20% < NDI < 40% and severe: NDI ≥ 40%)[22]. See Table 1 for descriptive statistics.

All participants provided written informed consent prior participation in the study and the study was approved by the Regional Ethical Review Board in Linköping (DNR 2011/262-32). All experiments were performed with the relevant guidelines and regulations.

## Inclusion and exclusion criteria

The WAD cohort was included from an ongoing randomized controlled trial (RCT) comparing three different exercise strategies; neck specific exercises, neck specific exercises in combination with a behavioral approach, or prescribed general physical activity [23, 24]. All participants in this study underwent the MR-scan before entering the RCT. The inclusion criteria for entering the RCT were: Age 18–62 years old, right handed with dominant right-sided pain, WAD > 6 months and < 3 years, Quebec Task Force WAD of grade II or III (grade II means neck complaints and musculoskeletal signs; grade III means grade II plus neurological signs [4]) and pain intensity of greater than 20 mm on a 100-mm visual analog scale [25] and/ or a score greater than 20% on the NDI scale).

Exclusion criteria from the WAD cohort were known or suspected serious physical pathology, earlier fracture or luxation of the cervical column, neck trauma with persistent symptoms from previous injury, surgery on the cervical column; neck pain that caused absence from work >1 month in the year prior to the WAD trauma, signs of traumatic brain injury from or before the whiplash injury, generalized or more dominant pain elsewhere in the body, diseases or other injuries that might prevent full participation in the study, diagnosis of a severe psychiatric disorder, known drug abuse, contradiction for MRI, or insufficient knowledge of the Swedish language to answer the questionnaires.

The age- and sex-matched healthy cohort was included for comparative purposes to the WAD cohort. The exclusion criteria were present or past neck pain, dysfunction, or related disability, history of neck trauma, or lower back pain, rheumatologic or neurological disease, generalized myalgia and institutional contraindications for undergoing an MRI exam.

## Magnetic resonance imaging

All images were acquired with a Philips Ingenia 3.0T scanner (Royal Philips, Amsterdam, the Netherlands). The coils used for imaging was the built-in phased array posterior coil, a 32-channel head coil and two anterior flexible coils.

*The neck images* were acquired using a 2-point Dixon 3D gradient-echo sequence with out-of-phase echo time of 3.66 milliseconds, and in-phase echo time of 7.24 milliseconds. TR was 10 milliseconds; the flip angle was 10° and the acquisition time was 9 minutes. Later echoes

enabled the acquisition of a high resolution of 0.75*0.75*0.75 mm$^3$. A high resolution is needed to distinguish the small muscles in the deep neck musculature. Literature values of T2* relaxation (23.9 milliseconds for water and fat) was used for T2* correction (23,24). One participant was excluded due to a fat-water swap artifact.

*The lower extremities* were imaged using a 2-point Dixon 3D gradient-echo sequence with out of phase echo time of 1.15 milliseconds, and in-phase echo time of 2.3 milliseconds. TR was 3.78 milliseconds and the flip angle was 10˚. The acquisition time was 5 minutes and the voxel size was 1.75*1.75*1.75 mm$^3$. Water- and fat-separated images were acquired using a phase-based reconstruction method [26]. In the lower extremities, an intensity inhomogeneity correction was performed using fat signal referencing [27]. The fat-referenced imaging technique has only been validated for short echo-times and was therefore not applied to the neck images. As for the neck images, literature values for T2* correction were applied (23, 24).

## Measurement of muscle fat infiltration

Left and right multifidi were segmented in the 3D volume from the level of cervical vertebra 3–7 using the semi-automatic image foresting transform technique [28]. Shortly described, the technique uses an algorithm that calculates a segmentation based on a few manually defined foreground seeds (pixels within the region of interest) and background seeds (outside the region of interest). This process is iterated by adding foreground and background seeds to help the algorithm until the operator is satisfied. The segmentation was done on the water image, and large fatty streaks adjacent to the muscle were excluded. The segmentation was performed by a musculoskeletal physiotherapist with more than 6 months experience doing this particular analysis. Fourteen randomly selected participants were segmented twice for investigating intra-rater reliability. The intra class correlation was 0.90 (CI 0.76–0.99) using two-way random-effects, single-measure intra-class correlation coefficient with absolute agreement. A physiotherapist researcher with more than 10 years of medical imaging experience also confirmed the segmentations. The MFI within the neck was acquired by calculating the fat signal fraction: $MFI\ (\%) = 100 \cdot \frac{fat}{fat+water}$.

The lower extremities were automatically segmented using a multi-atlas segmentation technique presented by Karlsson et al. [29]. The method can handle different field strengths (1.5 T and 3.0 T) and image resolutions [29, 30] and has also been validated using test-retest with lean, overweight and obese participants [31]. All muscles below the hip joint were included. Fifteen atlases with annotated muscles were non-rigidly registered onto the subject followed by majority voting, i.e. more than half of the atlases needed to agree to classify the tissue as muscle. MFI was defined as the average fat in muscle tissue where the fat concentration was less than 50%. This way, contributions of fat signal from large fatty streaks and potential leakage in subcutaneous adipose tissue were avoided. The range of limits of agreement for the precision of this MFI measure is $<\pm 2\%$ for leg muscles [30].

## Statistics

The Shapiro-Wilk's test together with visual inspection of histograms were used for investigating if MFI in the neck and lower extremities were approximately normally distributed. Although the Schapiro-Wilk's test could not exclude non-normal distribution, the visual inspection of the histograms and box plots showed that MFI in multifidi and the lower extremities respectively still could be considered as normally distributed.

The analysis was performed using a mixed linear model with (A)—the association between MFI in multifidi with MFI in the lower extremities, (B)–model A with WAD-group and age as fixed factors, and (C)–model B with BMI as an additional factor. A linear regression between

multifidi and duration since injury was also made on the two WAD groups. Analysis of variance (ANOVA) was used to investigate if there are any differences in MFI in the lower extremities between severe WAD, mild/moderate WAD and healthy controls. A Bonferroni correction of multiple comparisons was chosen as post-hoc test to investigate potential between-group differences. A p-value < 0.05 was considered significant.

## Results

A typical result for the semi-automatic segmentation of the neck is showed in Fig 1 and Fig 2 shows a typical example of the fully automatic segmentation of the lower extremities.

There was a significant linear association (p<0.001) between MFI in multifidus and MFI in the lower extremities (Model A) with $r^2$ = 0.28. This relation remained significant when the model was adjusted for age, and WAD-group (Model B, p = 0.009) as well as when adjusted for age, WAD-group and BMI (Model C, p = 0.002). The severe WAD group had a significant (p = 0.0016) and high correlation ($r^2$ = 0.69) between MFI in multifidi and MFI in the lower extremities. The healthy controls also showed a significant (p = 0.022) correlation ($r^2$ = 0.17). However, mild/moderate WAD showed no significant (p = 0.29) correlation between MFI in multifidus and MFI in the lower extremities ($r^2$ = 0.06). Fig 3 shows the MFI in multifidi plotted against the MFI in the lower extremities color coded for the different groups. Observe that caution should be made for comparing the $r^2$ from the different subgroups due to different within-group variances. WAD group was a significant factor both in model B (p = 0.017) and in model C (p = 0.024) when comparing multifidus MFI to MFI in the lower extremities. Severe WAD had significantly higher slope than both mild/moderate WAD(model B, p = 0.005, model C, p = 0.01) and healthy controls (model B, p = 0.016, and model C, p = 0.015). Also, age was a significant factor both in model B (p = 0.009) and in model C (p = 0.008). BMI did not show as a significant factor in model C (p = 0.065). There was no significant (p = 0.9) association between MFI in multifidi and duration since injury.

The mean MFI in the lower extremities was 5.04% for the healthy controls, 5.06% for the mild/moderate WAD and 6.25% for the severe WAD. The ANOVA showed no statistically significant differences (p = 0.11) in the lower extremities' MFI between the different groups. Fig 4 shows the mean MFI in the lower extremities for the three different groups.

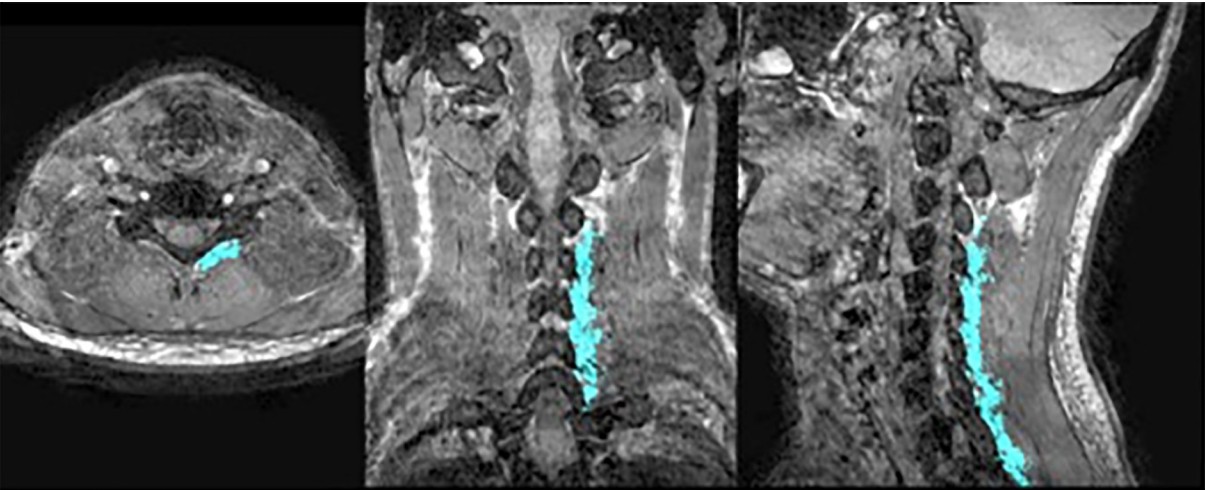

**Fig 1. Segmentation of m. multifidi.** An axial (left), a coronal (middle), and a sagittal (right) cut of a high-resolution (0.75*0.75*0.75 mm$^3$) 3D image volume acquired using 2-point Dixon imaging followed by water-fat separation highlighting the segmentation of the left cervical multifidii in one of the healthy controls. The segmented mask is overlaid on the fat + water image volume.

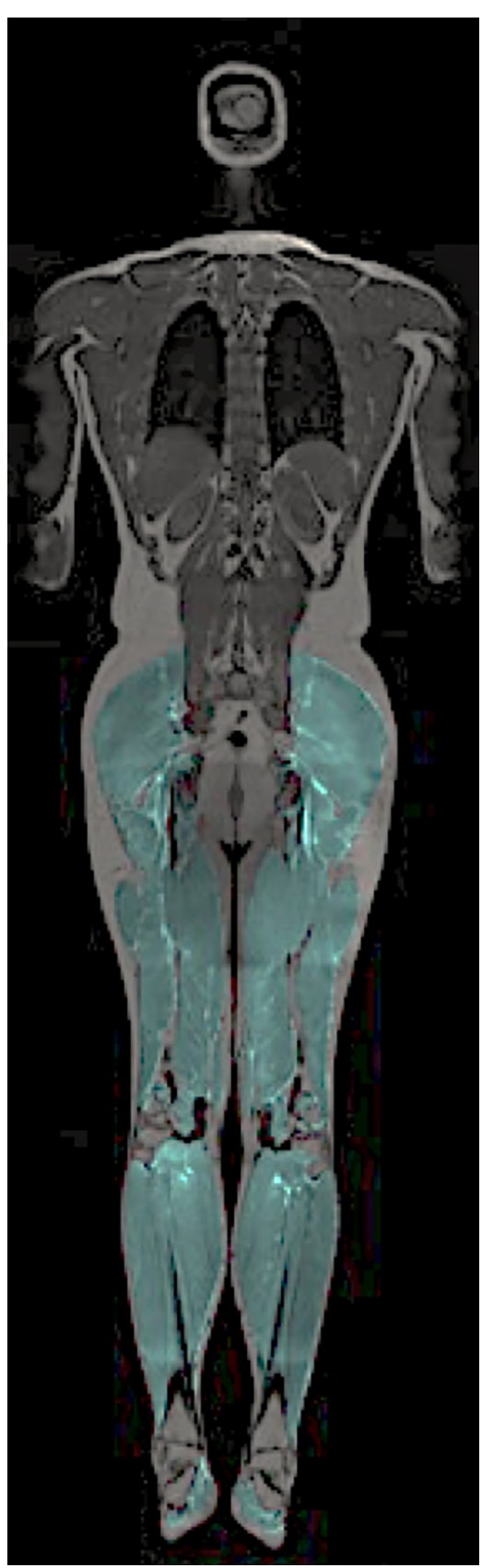

**Fig 2. Segmentation of lower extremities.** A coronal slice of the selected distal muscles (lower extremities) that was included in this study. The distal muscles were automatically defined using multi-atlas segmentation and is overlaid the water-fat images in a different color.

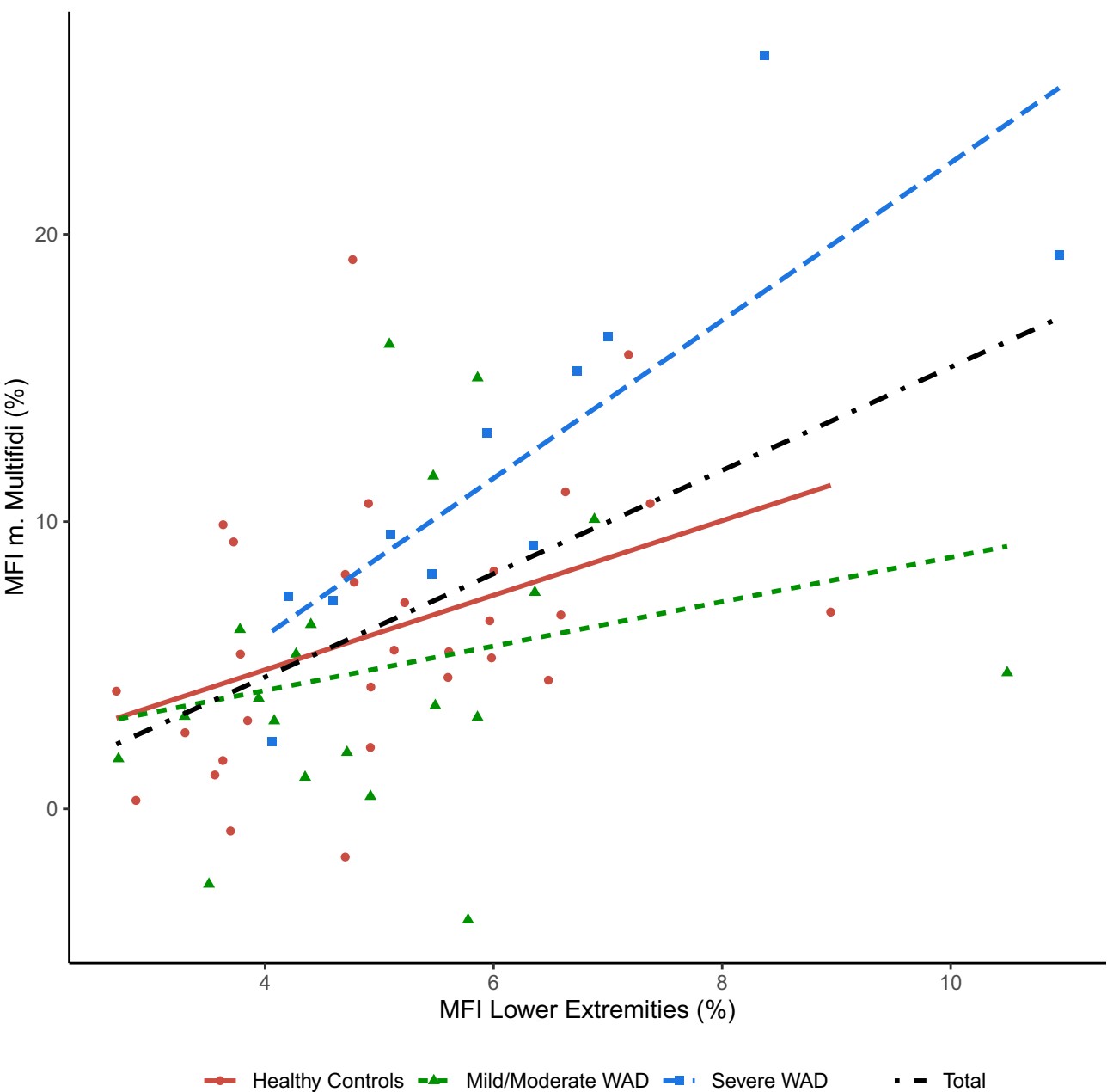

**Fig 3. Association between MFI in multifidi and lower extremities.** The MFI of the m. multifidi plotted against MFI in the lower extremities using different colors and shapes for the different WAD groups. The MFI is reported as the fraction of fat in the muscle (%). Regression lines are fitted to the data: Total (Black, dot dashed), y = -2.6 + 1.8x, $r^2$ = 0.28, p < 0.001; Healthy Controls (Red, solid), y = -0.35 + 1.3x, $r^2$ = 0.17, p = 0.022; Mild/Moderate WAD (green, short dashed), y = 1.0 + 0.77x, $r^2$ = 0.29, p = 0.29; Severe WAD (blue, long dashed), y = -5.0+2.7x, $r^2$ = 0.69, p = 0.0016.

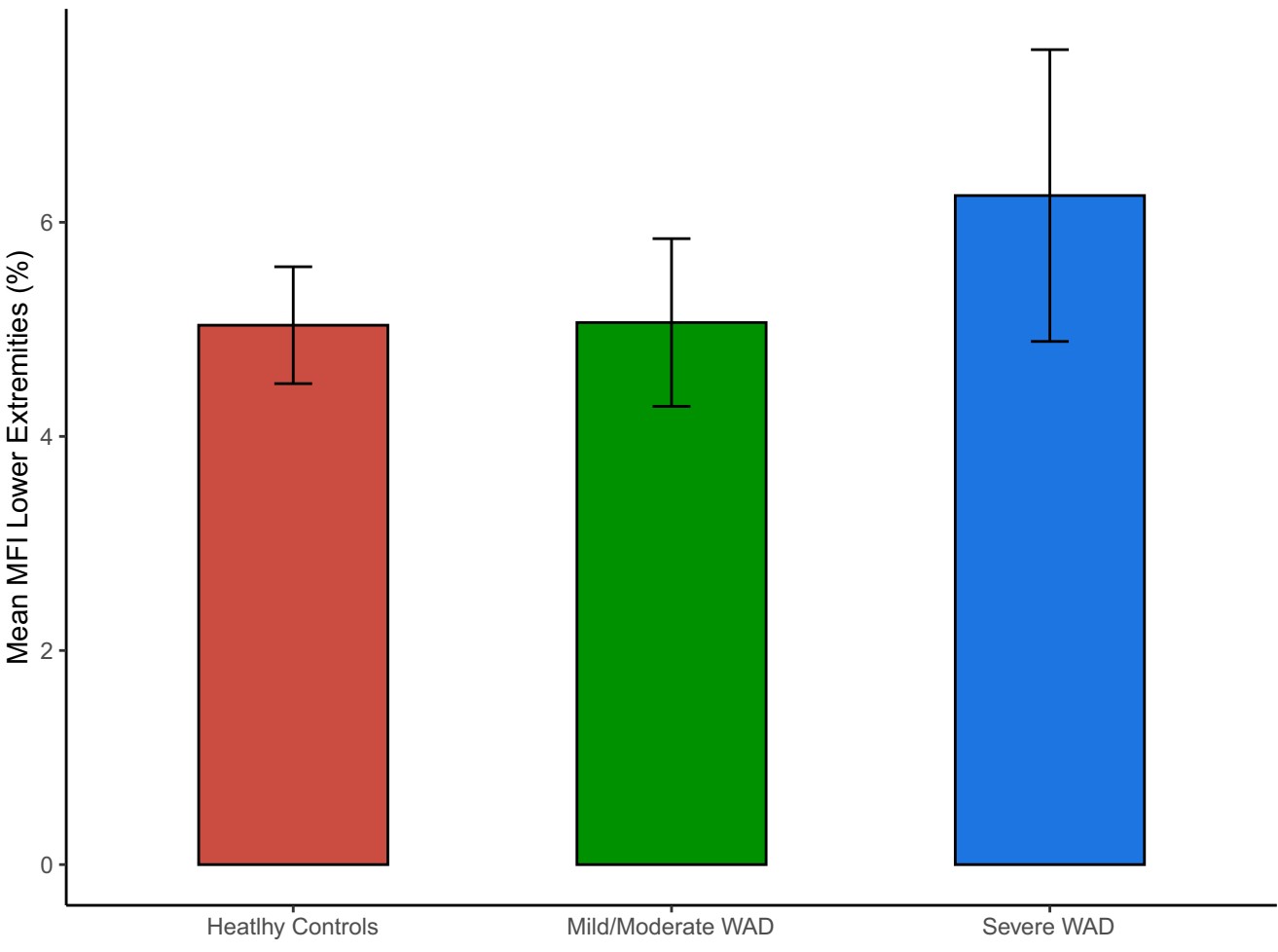

**Fig 4. Mean MFI in the three different groups.** The figure shows the MFI based on the volumetric analysis of the lower extremities. The top of the bars represents the mean MFI value (in %) and the whiskers show the 95% confidence intervals.

## Discussion

There was a significant linear association between the MFI in the multifidi muscles and in the lower extremities with $r^2$ = 0.28 (model A). This association remained significant when adding age and WAD group (model B) as well as BMI (model C) as potential confounding factors, suggesting that self-reported disability after WAD injury and MFI distal to the whiplash trauma might be linked. This potential link is further implied when looking at the correlations of each group. Severe WAD shows a strong correlation between multifidi MFI and MFI in the lower extremities, with a $r^2$ = 0.69. Furthermore, mild/moderate WAD shows no significant correlation between local and distal MFI.

The duration since injury is not correlated to the MFI in multifidi. Patients with severe WAD had significantly higher MFI in multifidi both in model B and model C, implying that self-reported disability and MFI are connected although the time since injury does not affect the result. One hypothesis is that a higher fat infiltration (or degenerative pathologies) prior the trauma might affect the recovery of the patient groups [32]. While it is largely unknown if and how changes in MFI is associated with poor functional recovery from whiplash, the results

of this study support that higher amounts of MFI could contribute to predictive models towards determining whiplash recovery or other common conditions. A previous study investigating fibromyalgia, found significantly higher MFI values in the quadriceps muscles compared to healthy controls [33]. However, note that widespread pain was an exclusion criterion in this study.

The difference in MFI in the lower extremities was not statistically significant between the groups according to the ANOVA. This is similar to the findings by Pedler et al where there were no significant differences between MFI in multifidi and the right soleus muscle [9]. These findings indicate complex mechanistic pathways influence MFI development and that physical inactivity may not play a significant role in the recovery after a whiplash trauma since generalized MFI was not significantly different between the groups. Nevertheless, the potential that higher MFI prior the collision or larger expressions following the collision may influence a systemic response needs to be studied in a larger inception cohort in longitudinal fashion. However, to draw such conclusions, prognostic studies are needed. Our findings do indicate that the whiplash trauma might have a direct or indirect effect on the muscles close to the cervical spine may be at play but not as featured to occur in distal muscles. Further mechanistic work involving a larger sample of participants with varying levels of signs and symptoms and across different muscles is required before definitive conclusions can be drawn.

A case-study also reported high MFI in both the neck region and the lower extremities compared to one recovered participant [15], supporting the heterogeneity of the whiplash injury and recovery thereof. There are some differences between the designs comparing that study from this. The inclusion criteria for the studies are not in alignment. Also, the case study [15] involved participants with a more complex chronic WAD and one recovered participant. While the severity of injury is largely unknown, future quantitative work investigating spinal cord pathways and heightened molecular–neuroimmune—responses across a number of patient populations is warranted, available [34–38] and underway to fully understand the mechanisms underlying the cause and progression of compositional whole-body muscle changes in WAD, and other conditions. In the present study there were no clinical or radiological signs of a known spinal cord injury (e.g. jumped facet joints), however minor insults involving the spinal cord cannot be excluded.

While the possibilities to measure MFI in the small muscles in the neck with high-resolution MRI continues to evolve, the wider literature would benefit from a clear and broadly accepted definition of MFI. Suggestions on those definitions are, for example, proposed in the study by Crawford et al. [39] and the study by Elliott et al. [40]. The measures should be reproducible and insensitive to different scanners, different laboratories, and performed using consensus driven methodologies. In addition, future research should investigate MFI across a number of common conditions to compare and contrast diagnosis-dependent changes in local and whole-body MFI. The prospects of training and using deep learning neural networks also increases the feasibility of translating such measures to clinical practice [41].

The Shapiro-Wilks test indicated that the distribution was slightly non-normal. A non-parametric analysis was therefore performed and confirmed the results from the parametric test. For future studies, larger sample sizes could permit the development of larger, more complex, models without losing statistical power. However, this study establishes a potentially new link with MRI whole-body acquisition, which is not a normal clinical routine for patients with suspected spine trauma following motor vehicle collision. These links between local and distal MFI are intriguing and are opening up for further studies of the distal MFI's influence to high self-reported disability in chronic WAD. With the evolved MRI technique, a head-to-knee protocol with enough resolution to automatically analyze the composition of the thigh muscles is possible using a six minute scan [42].

One limitation with this study was that no functional analysis was made on the lower extremities. Furthermore, no exclusion was made for previous trauma or injury in the lower extremities. However, an exclusion criterion for this study was dominating or generalized pain (other than for the neck region for WAD participants) minimizing the potential influences of previous trauma/injury.

Another limitation with this study was that the methods for measuring MFI in multifidi and lower extremities differs slightly. In the lower extremities, lower resolution is sufficient, which enables a shorter MR scan. Furthermore, the muscles can be analyzed with automatic methods with high reported precision [29, 30]. With evolving MRI technology better possibilities for scanning the smallest muscles (e.g. cervical multifidi) with enough resolution for segmentation and analysis have been possible [41]. In this manuscript the whole multifidi muscle volume has been analyzed instead of a few cross-sectional slices, which has been considered the previous gold standard. When the entire muscle is analyzed the result is not sensitive to the placement of single image slices. This increased precision makes it easier to perform longitudinal studies investigating e.g. different rehabilitation programs or to follow the progression of the WAD condition. However, one limitation with our approach is that a single peak lipid model with a theoretical value for $T2^*$ was used. This could induce a $T2^*$ bias in the investigation. Another limitation is that no calibration of the fat signal was applied to the high-resolution neck images. This may have introduced T1-bias in calculating the fat content of the m. multifidi. Future works including technical development of quantitative MFI measurement in the small deep neck muscles are of high interest to further investigate the local changes in MFI after, not only a whiplash injury, but other common, yet equally enigmatic, degenerative and pathological conditions of the spine (e.g. myelopathy and radiculopathy).

To conclude, since there was no significant difference in MFI distal to the trauma in the severe WAD group compared to either the mild/moderate WAD group, or to the healthy group, the results indicate the response to the whiplash trauma reflects a local physiological process in patients with severe chronic WAD. Furthermore, time since injury did not have a significant correlation with MFI in the multifidi, which indicate that slow development of generalized MFI due to e.g. physical inactivity after the trauma appears to have no major effect. Finally, the strong association between multifidi MFI and MFI in the lower extremities found to be unique to the group with severe WAD ($r^2 = 0.69$) but not in the mild/minor WAD group ($r^2 = 0.06$) does not exclude the possibility that high generalized MFI prior the accident could play a role in the development of severe chronic WAD. The associations between MFI and WAD are intriguing and may in the future contribute to better understanding regarding onset and progress of WAD after a whiplash trauma.

## Supporting information

**S1 Table. Complete measurement results for all participants included in this study.** (XLSX)

## Author Contributions

**Conceptualization:** Anette Karlsson, Anneli Peolsson, Olof Dahlqvist Leinhard.

**Data curation:** Anette Karlsson, Anneli Peolsson, Thobias Romu, Helena Ljunggren, Olof Dahlqvist Leinhard.

**Formal analysis:** Anette Karlsson, James Elliott, Magnus Borga, Olof Dahlqvist Leinhard.

**Funding acquisition:** Anneli Peolsson, Olof Dahlqvist Leinhard.

**Investigation:** Anette Karlsson.

**Methodology:** Anette Karlsson, Thobias Romu, Olof Dahlqvist Leinhard.

**Project administration:** Anneli Peolsson.

**Software:** Anette Karlsson.

**Supervision:** Anneli Peolsson, Magnus Borga.

**Validation:** James Elliott.

**Visualization:** Anette Karlsson.

**Writing – original draft:** Anette Karlsson.

**Writing – review & editing:** Anette Karlsson, Anneli Peolsson, James Elliott, Thobias Romu, Helena Ljunggren, Magnus Borga, Olof Dahlqvist Leinhard.

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
