## [Decision Letter · Decision Letter 0]

7 Oct 2019

PONE-D-19-19642

The relation between local and distal muscle fat infiltration in chronic whiplash using magnetic resonance imaging

PLOS ONE

Dear Ms Karlsson,

Thank you for submitting your manuscript to PLOS ONE. After careful consideration, we feel that it has merit but does not fully meet PLOS ONE’s publication criteria as it currently stands. Therefore, we invite you to submit a revised version of the manuscript that addresses the points raised during the review process.

We would appreciate receiving your revised manuscript by Nov 21 2019 11:59PM. To enhance the reproducibility of your results, we recommend that if applicable you deposit your laboratory protocols in protocols.io, where a protocol can be assigned its own identifier (DOI) such that it can be cited independently in the future. For instructions see: http://journals.plos.org/plosone/s/submission-guidelines#loc-laboratory-protocols

We look forward to receiving your revised manuscript.

Kind regards,

Xi Chen

Academic Editor

PLOS ONE

Journal Requirements:

1. Thank you for including your competing interests statement; "We have read the journal's policy and the authors of this manuscript have the following competing interests: TR, MB, and ODL receive salaries and are stockholders of Advanced MR Analytics AB. AK is a stockholder of Advanced MR Analytics AB. This does not alter our adherence to PLOS ONE's policies on sharing data and material."

We note that one or more of the authors received salaries from a commercial company: Advanced MR Analytics AB

Reviewers' comments:

Reviewer's Responses to Questions

**Comments to the Author**

1. Is the manuscript technically sound, and do the data support the conclusions?

Reviewer #1: Yes

Reviewer #2: Yes

2. Has the statistical analysis been performed appropriately and rigorously? 

Reviewer #1: No

Reviewer #2: Yes

3. Have the authors made all data underlying the findings in their manuscript fully available?

Reviewer #1: Yes

Reviewer #2: No

4. Is the manuscript presented in an intelligible fashion and written in standard English?

Reviewer #1: Yes

Reviewer #2: Yes

5. Review Comments to the Author

Reviewer #1: I thank the authors for their work in addressing my comments and believe the paper is now much more focused and streamlined. I have a few additional comments and suggestions primarily related to the discussion of the data and conclusions drawn.

1. Introduction: I am still struggling with the connection between this data and the idea of “complementing the neck images with whole body coverage” for diagnostic purposes. First, you are unable to determine whether the stronger correlation between multifidi and lower extremity MFI in the severe WAD group pre-exists and predicts a worse outcome or is a progressive response to injury. The data that you discuss more strongly support the latter, in which case diagnostic whole body scans would not be useful. Second, to argue for whole body scans, you would have to show that the relationship between multifidi and lower extremity MFI is a better indicator than multifidi MFI on its own. This can be grossly assessed by adding the two tests discussed in the next points.

2. Results: Please perform an ANCOVA on your three regressions to determine whether the slopes are significantly different.

3. Results: Please also include an ANOVA on the multifidi MFI between groups to complement Figure 4.

4. Results: Please report all statistical results in your Results section. The Discussion discusses associations between multifidi MFI and age, BMI and duration since injury and a comparison of multifidi MFI across groups. If these are outcomes of the statistical models, please include more details about the models and their outputs so the reader is aware of the results before the discussion.

5. Discussion: Please discuss additional explanations for the association between multifidi and lower extremity MFI. These are mentioned in the Introduction, but I would like to see a more in-depth treatment in the Discussion. “However, a number of hypotheses around injury severity, pain intensity and/or related disability (9, 10), heightened stress-responses (11-13) central/peripheral neuronal interference (14-16), and/or physical inactivity (17, 18) may be offered to explain the rapid expression and larger magnitude of MFI in patients with more severe self-reported symptoms.”

Reviewer #2: Thank you for the opportunity to review this interesting manuscript for PLOS ONE. Our author colleagues have undertaken a study examining the relationship between cervical multifidus MFI and lower limb MFI in controls, mild WAD and mod-severe WAD patients. Findings indicate a relationship between cervical muscle MFI and lower limb MFI in mod-severe WAD patients.

General comments:

This study provides important insight on the mechanisms underpinning chronic WAD, which I believe represents an advance in our knowledge of this challenging condition.

I see that the authors have already made substantial amendments to the manuscript in response to two prior reviewers’ comments. My impression is that they have adequately addressed prior reviewer concerns.

Specific comments:

Abstract

1) “…to investigate the relation between fat infiltration…” awkward wording, suggest revise to ‘…relationship between…’

Introduction

2) Minor typographic error: “…whiplash from an motor vehicle collision.” Correct to ‘…a motor vehicle…’

3) Paragraph 2: “…reporting lower levels of pain-related disability, those nominating full recovery, idiopathic neck pain, and healthy controls.” Suggest add citation to this paper here: Elliott et al. (2008). Fatty infiltrate in the cervical extensor muscles is not a feature of chronic, insidious-onset neck pain. Clin Radiol. 63, 681-687

4) Paragraph 3: In line with the hypothesis around SCI in WAD, I suggest also mentioning the preliminary publication detailing magnetic resonance spectroscopic evidence of cord injury below, as a complementary study supporting this theory:

Elliott JM, Pedler AR, Cowin G, Sterling M, McMahon K. Spinal cord metabolism and muscle water diffusion in whiplash. Spinal Cord. 2012 Jun;50(6):474-6

Methods

5) “…technique uses foreground and backgrounds seeds and let the segmented mask grow.” Awkward wording, suggest revise.

6) “The segmentation was performed by a musculoskeletal physiotherapist with >6 months experience.” Suggest slight re-wording for clarity that it is 6 months experience doing this particular analysis (or as applicable).

7) Statistics: Was there a power/sample size calculation undertaken?

Discussion

8) The Discussion reads well. My only comment is that it would be good to address the contrasting findings between this study and that of Pedler et al (2018) which did not find increased lower limb MFI in WAD:

Pedler A, McMahon K, Galloway G, Durbridge G, Sterling M (2018) Intramuscular fat is present in cervical multifidus but not soleus in patients with chronic whiplash associated disorders. PLoS ONE 13(5): e0197438. https://doi.org/10.1371/journal.pone.0197438

6. PLOS authors have the option to publish the peer review history of their article (what does this mean?). If published, this will include your full peer review and any attached files.

Reviewer #1: No

Reviewer #2: No

---

## [Author Response · Author response to Decision Letter 0]

31 Oct 2019

Journal Requirements:

 We have adapted the manuscript to PLOS ONE's style requirements. 

1. Thank you for including your competing interests statement; "We have read the journal's policy and the authors of this manuscript have the following competing interests: TR, MB, and ODL receive salaries and are stockholders of Advanced MR Analytics AB. AK is a stockholder of Advanced MR Analytics AB. This does not alter our adherence to PLOS ONE's policies on sharing data and material."

We note that one or more of the authors received salaries from a commercial company: Advanced MR Analytics AB

FUNDING STATEMENT 

"The study received funding from the Swedish Research Council and the Medical Research Council of South-East Sweden (FORSS). The funders had no role in study design, data collection and analysis, decision to publish, or preparation of the manuscript. Furthermore, the commercial company AMRA Medical AB provided support in the form of salaries for authors TR, MB and ODL, but did not have any additional role in the study design, data collection and analysis, decision to publish, or preparation of the manuscript. The specific roles of these authors are articulated in the ‘author contributions’ section."

COMPETING INTEREST STATEMENT

We have read the journal's policy and the authors of this manuscript have the following competing interests: TR, MB, and ODL receive salaries and are stockholders of AMRA Medical AB. AK is a stockholder of AMRA Medical AB. This does not alter our adherence to PLOS ONE's policies on sharing data and material.

SPECIFIC COMMENTS TO REVIEWER ARE FOUND IN THE SUPPORTING MATERIAL BUT ALSO CUT IN HERE: 

Reviewer #1: 

I thank the authors for their work in addressing my comments and believe the paper is now much more focused and streamlined. I have a few additional comments and suggestions primarily related to the discussion of the data and conclusions drawn.

OUR ANSWER: We thank you for once again taking the time to provide insightful comments to further improve our manuscript. 

1. Introduction: I am still struggling with the connection between this data and the idea of “complementing the neck images with whole body coverage” for diagnostic purposes. First, you are unable to determine whether the stronger correlation between multifidi and lower extremity MFI in the severe WAD group pre-exists and predicts a worse outcome or is a progressive response to injury. The data that you discuss more strongly support the latter, in which case diagnostic whole-body scans would not be useful. Second, to argue for whole body scans, you would have to show that the relationship between multifidi and lower extremity MFI is a better indicator than multifidi MFI on its own. This can be grossly assessed by adding the two tests discussed in the next points.

OUR ANSWER: We apologize for unclear phrasing in the introduction. We understand the interpretation of us claiming inclusion of whole-body scans for diagnostic purposes, but this was not our intention. It was rather to communicate that studies involving whole-body scans can provide valuable information and knowledge in order to understand the mechanism behind fat infiltration in the multifidi muscles and why some individuals never fully recover from a whiplash trauma. We have now revised parts of the introduction to avoid the risk of such misinterpretation. We have also re-written the statistic section, the result and the discussion to bring more clarity regarding the suggested tests, please see our responses on points 2-4 below. 

2. Results: Please perform an ANCOVA on your three regressions to determine whether the slopes are significantly different.

OUR ANSWER: In this study we used three different mixed linear models. In models B and C, group was included as a factor. In both these models, group differences were significant. This was unfortunately not clearly stated in the result section earlier. We have also made some changes in the discussion, conclusion and in our abstract based on comment 1-3 where we also emphasize the implications of a local injury more prominently than in the earlier version of this manuscript. 

3. Results: Please also include an ANOVA on the multifidi MFI between groups to complement Figure 4.

OUR ANSWER: An ANOVA was already performed to complement Figure 4. We changed the wording in the result section to present it more distinctly. 

Changes made: We replaced “The difference in the lower extremities’ MFI between the groups was not statistically significant (p = 0.11)” with “The ANOVA showed no statistically significant differences (p=0.11) in the lower extremities’ MFI between the different groups”

4. Results: Please report all statistical results in your Results section. The Discussion discusses associations between multifidi MFI and age, BMI and duration since injury and a comparison of multifidi MFI across groups. If these are outcomes of the statistical models, please include more details about the models and their outputs so the reader is aware of the results before the discussion.

OUR ANSWER: We realize that we have lacked in clarity in presenting our statistical analyses. In the statistic part of the method section we have addressed the different statistical models including ANOVA of the group measurements and the three different variations of the mixed linear models, which included variations of WAD group, age, BMI. A test of MFI versus Duration in the two WAD groups is also explained in that section. We have also improved consistency regarding the wording throughout the method, result and discussion to minimize miscommunication with a potential reader regarding which statistical analyses that have been performed. 

5. Discussion: Please discuss additional explanations for the association between multifidi and lower extremity MFI. These are mentioned in the Introduction, but I would like to see a more in-depth treatment in the Discussion. “However, a number of hypotheses around injury severity, pain intensity and/or related disability (9, 10), heightened stress-responses (11-13) central/peripheral neuronal interference (14-16), and/or physical inactivity (17, 18) may be offered to explain the rapid expression and larger magnitude of MFI in patients with more severe self-reported symptoms.”

OUR ANSWER: We have added a reference suggested by reviewer 2 and also extended the discussion (paragraph 3 in the discussion) with a more detailed discussion of what the findings of this study might imply. 

Reviewer #2: 

Thank you for the opportunity to review this interesting manuscript for PLOS ONE. Our author colleagues have undertaken a study examining the relationship between cervical multifidus MFI and lower limb MFI in controls, mild WAD and mod-severe WAD patients. Findings indicate a relationship between cervical muscle MFI and lower limb MFI in mod-severe WAD patients.

General comments:

This study provides important insight on the mechanisms underpinning chronic WAD, which I believe represents an advance in our knowledge of this challenging condition.

I see that the authors have already made substantial amendments to the manuscript in response to two prior reviewers’ comments. My impression is that they have adequately addressed prior reviewer concerns.

OUR ANSWER: Thank you for your time and suggestion on how to improve this manuscript further. Below you will find our responses to the comments. 

Specific comments:

Abstract

1) “…to investigate the relation between fat infiltration…” awkward wording, suggest revise to ‘…relationship between…’

OUR ANSWER: Thank you. We have changed the wording in the abstract as suggested. 

Introduction

2) Minor typographic error: “…whiplash from an motor vehicle collision.” Correct to ‘…a motor vehicle…’

OUR ANSWER: Thank you for noting the error. It has been corrected. 

3) Paragraph 2: “…reporting lower levels of pain-related disability, those nominating full recovery, idiopathic neck pain, and healthy controls.” Suggest add citation to this paper here: Elliott et al. (2008). Fatty infiltrate in the cervical extensor muscles is not a feature of chronic, insidious-onset neck pain. Clin Radiol. 63, 681-687

OUR ANSWER: The suggested article is now cited in the manuscript at the suggested location in paragraph 2. 

4) Paragraph 3: In line with the hypothesis around SCI in WAD, I suggest also mentioning the preliminary publication detailing magnetic resonance spectroscopic evidence of cord injury below, as a complementary study supporting this theory:

Elliott JM, Pedler AR, Cowin G, Sterling M, McMahon K. Spinal cord metabolism and muscle water diffusion in whiplash. Spinal Cord. 2012 Jun;50(6):474-6

OUR ANSWER: We added the potential effect of spinal cord injury with reference to this preliminary study in the third paragraph in introduction section. 

Methods

5) “…technique uses foreground and backgrounds seeds and let the segmented mask grow.” Awkward wording, suggest revise.

OUR ANSWER: We have revised the sentence to remove the awkward wording and to improve clarity. 

New sentence: “Shortly described, the technique uses an algorithm that calculates a segmentation based on a few manually defined foreground seeds (pixels within the region of interest) and background seeds (outside the region of interest).”

6) “The segmentation was performed by a musculoskeletal physiotherapist with >6 months experience.” Suggest slight re-wording for clarity that it is 6 months experience doing this particular analysis (or as applicable).

OUR ANSWER: This has now been clarified in the method section. 

This is the new sentence: “The segmentation was performed by a musculoskeletal physiotherapist with more than 6 months experience doing this particular analysis.”

7) Statistics: Was there a power/sample size calculation undertaken?

OUR ANSWER: No, there was not. At the time for applying ethical approval and starting of the data acquisition, no reference literature measuring fat infiltration using fat- and water separation with age- and gender matched controls were found. In addition, literature has found indications that fat infiltration plays a role in WAD but estimating the smallest effect size that would be of scientific interest is still hard. However, we hope that with the evolved techniques around muscle fat infiltration with MR and the results from this study and others (e.g. Pedler et al 2018) will help in starting future studies with larger number of participants for further understanding of the mechanism of the whiplash trauma. 

Discussion

8) The Discussion reads well. My only comment is that it would be good to address the contrasting findings between this study and that of Pedler et al (2018) which did not find increased lower limb MFI in WAD:

Pedler A, McMahon K, Galloway G, Durbridge G, Sterling M (2018) Intramuscular fat is present in cervical multifidus but not soleus in patients with chronic whiplash associated disorders. PLoS ONE 13(5):https://doi.org/10.1371/journal.pone.0197438

OUR ANSWER: Thank you for the suggestion. We have addressed the study by Pedler et. al both in the introduction (last paragraph before the aim) and in the discussion (paragraph 3) to help the potential reader to understand not only the contrasting findings but also the similarities in the findings.

---

## [Decision Letter · Decision Letter 1]

19 Nov 2019

The relation between local and distal muscle fat infiltration in chronic whiplash using magnetic resonance imaging

PONE-D-19-19642R1

Dear Dr. Karlsson,

We are pleased to inform you that your manuscript has been judged scientifically suitable for publication and will be formally accepted for publication once it complies with all outstanding technical requirements.

With kind regards,

Xi Chen

Academic Editor

PLOS ONE

Additional Editor Comments (optional):

Reviewers' comments:

Reviewer's Responses to Questions

**Comments to the Author**

1. If the authors have adequately addressed your comments raised in a previous round of review and you feel that this manuscript is now acceptable for publication, you may indicate that here to bypass the “Comments to the Author” section, enter your conflict of interest statement in the “Confidential to Editor” section, and submit your "Accept" recommendation.

Reviewer #2: All comments have been addressed

Reviewer #3: (No Response)

2. Is the manuscript technically sound, and do the data support the conclusions?

Reviewer #2: Yes

Reviewer #3: (No Response)

3. Has the statistical analysis been performed appropriately and rigorously? 

Reviewer #2: Yes

Reviewer #3: (No Response)

4. Have the authors made all data underlying the findings in their manuscript fully available?

Reviewer #2: Yes

Reviewer #3: (No Response)

5. Is the manuscript presented in an intelligible fashion and written in standard English?

Reviewer #2: Yes

Reviewer #3: (No Response)

6. Review Comments to the Author

Reviewer #2: Thank you to the authors for their revisions of the manuscript. They have appropriately addressed the comments that I raised in my earlier review.

My only minor comment is that there is a typographic error in the spelling of Shapiro in the first paragraph of the Statistics section of the Method.

Reviewer #3: Previous reviewer's comments were fully addressed by the authors. It has significant improvement over the previous version and would be suitable for publication in the current form.

7. PLOS authors have the option to publish the peer review history of their article (what does this mean?). If published, this will include your full peer review and any attached files.

Reviewer #2: Yes: Scott Farrell PhD

Reviewer #3: No

---

## [Editor Report · Acceptance letter]

25 Nov 2019

PONE-D-19-19642R1 

The relation between local and distal muscle fat infiltration in chronic whiplash using magnetic resonance imaging 

Dear Dr. Karlsson:

I am pleased to inform you that your manuscript has been deemed suitable for publication in PLOS ONE. Congratulations! Your manuscript is now with our production department. 

With kind regards,

on behalf of

Dr. Xi Chen 

Academic Editor

PLOS ONE